# OpenReview forum: "Emergent Misalignment is Easy, Narrow Misalignment is Hard"
_ICLR.cc/2026/Conference — ICLR 2026 Poster_

### Official Review · Reviewer_X7FP · 2025-10-30

**Soundness:** 3
**Presentation:** 2
**Contribution:** 3
**Rating:** 6
**Confidence:** 4

**Summary:**

The paper investigates the phenomenon of emergent misalignment (EM) in LLMs, where finetuning on a narrow set of harmful examples leads to broad, stereotypical misaligned behavior across diverse, unrelated settings. The central contribution is an investigation into the inductive biases that favor this general misaligned solution over a narrow, task-specific one. The authors find that the general solution is demonstrably more efficient, stable, and influential in the pretraining distribution compared to the narrow solution, which they successfully enforce using a KL divergence loss. Overall, the paper is highly interesting, and its findings provide significant insights into the generalization and alignment properties of LLMs.

**Strengths:**

The originality of this work stems from its novel framing of emergent misalignment as a problem of competing solutions—general versus narrow—for a localized finetuning task. The work isolates and compares concrete linear representations for both the general and narrow solutions within the model's activation space, which is a powerful and highly original methodological approach to studying inductive biases. The quality of the empirical evidence is high, demonstrating the robustness of the EM phenomenon across different datasets, model scales, and architectures. The analysis comparing the efficiency, stability, and influence of the two solution types provides a clear, quantitative explanation for the observed inductive bias, which significantly advances our understanding of generalization in LLMs. The clarity of the core findings is strong. The significance is substantial, as it offers a foundational case study for monitoring and potentially mitigating future, more complex forms of emergent misalignment.

**Weaknesses:**

The analysis of the narrow misalignment solution, which is achieved via the introduction of a KL divergence constraint, introduces both practical and theoretical limitations. The success of this method critically relies on the assumption that the base pre-trained model is perfectly aligned. If the base model contains subtle, unknown, or slight misalignments, the KL penalty could inadvertently lock in these undesirable biases, thereby preventing the realization of a truly safe and narrowly-focused behavior. Furthermore, the practical deployment of this technique is hampered by the significant hyperparameter tuning challenge of balancing the KL divergence weight $\beta$ against the task-specific loss. The paper successfully demonstrates the existence of the narrow solution but does not offer a theoretically robust framework or an automatically adaptable mechanism for setting the $\beta$ value across different tasks or models.

A second major weakness concerns the explanatory power of the linear representation analysis. While the finding that both general and narrow misalignments can be captured by linear directions is extremely useful, the core arguments about the inductive bias being driven by the linear solution’s efficiency and stability are based on an approximation of a fundamentally non-linear process. The comparison of linear projections, while insightful, ultimately explains what happened in the model's feature space, but it does not fully uncover the deeper non-linear mechanisms responsible for organizing the pretraining feature space in a way that makes the linear general solution globally optimal or locally most efficient for the fine-tuning loss.

Finally, the overall clarity and presentation of the paper could be significantly improved. The prose is occasionally non-fluent, requiring careful editing for an ICLR submission. Specifically, Figure 3, which currently appears to be a screenshot or a less polished visualization, requires professional reformatting to better present the complex comparison data it contains, which would enhance the overall readability of the results.

**Questions:**

Could the authors elaborate on the stability of the narrow misalignment solution (NM) with respect to different values of the KL divergence weight $\beta$? Specifically, is there a phase transition where a small change in $\beta$ causes the model to jump from an NM solution to an EM solution, and what does this imply for the practical robustness of the NM solution?

The paper argues the general solution is more influential in the pre-training distribution. Can the authors provide a more detailed information-theoretic or geometric explanation as to why the pre-training objective would favor the encoding of a general concept that is so easily activated by a narrow, harmful dataset? What specific properties of the pre-training data or objective create this specific inductive bias?

Please ensure that all figures, particularly Figure 3, are redrawn and polished to meet the professional standards of a conference submission, ensuring a more fluid reading experience. The overall prose requires a pass for improved fluency and clarity.

---

> ### Author Response · Authors · 2025-11-16
> **Response to X7FP**
>
> Thank you for your engagement and detailed review of our work! We appreciate your acknowledgement of the clarity and substantial significance of our findings as a foundational case-study in this area. We also appreciate the thoughtful suggestions and questions raised, which we address below and have incorporated into the revised PDF to strengthen the paper.
>
> **Regarding challenges with practical deployment.** We appreciate your acknowledgement that the KL regularisation approach has practical applications, and agree that overcoming misalignment in the base model is a major challenge to be overcome, by this and other finetuning approaches. However, to clarify, we see the main contribution of our work is to study generalisation in finetuning, using EM as a valuable mechanistic case-study to advance our scientific understanding. We do not immediately aim to propose practical mitigations to misalignment, so have not focused on addressing practical aspects related to our method’s deployment.
>
> **Regarding the analysis of linear representations.** In our narrow and general misalignment experiments in Section 3, we study both steering vectors and LoRA adapters. We find that, as for the linear steering vectors, the general misalignment LoRA adapters show greater efficiency and stability than their narrowly misaligned counterparts. We also find that the non-linear LoRA and all-parameter finetunes are very well approximated by the linear solutions, as evidenced by our finetuning results and ablation results, which shows that a linear intervention can mediate the behaviour of the non-linear finetuning processes. Given this, we believe that our results do offer valuable insights into the core question of why general rather than narrow misalignment is learnt, despite not characterising the full non-linear pretraining feature space.
>
> **Impact of KL regularization weight.** Thank you for this interesting question about the KL regularisation dynamics. We find that as the KL penalty is increased, there is a gradual decrease in general misalignment. If it is increased significantly above the paper’s selected value of 1e-6, the finetuning is constrained such that we no longer observe narrow or general misalignment. We have incorporated this result into Appendix H.
>
> **Explaining pre-training preferences.** We strongly agree that the question of how pre-training shapes inductive biases is a fascinating question. However, unfortunately a formal explanation of how concepts become encoded in pretraining is out of scope for this work, as is running expensive pretraining experiments to determine how such concepts are shaped by the data or training objectives.
>
> **Figures and clarity.** We greatly appreciate your assistance in improving the clarity of the work. We have replaced Figure 3 with a Table in order to enhance readability, and have increased the clarity of key points, such as the distinction between narrow and general misalignment, as described in our responses to the other reviewers.
>
> We greatly appreciate your feedback and engagement in helping us improve our work, and would be delighted to answer any further questions.

---

> > ### Comment · Reviewer_X7FP · 2025-11-26
> > **Official Reply to the Authors**
> >
> > I thank the authors for their detailed response and this insightful work. And I will maintain my score.

---

### Official Review · Reviewer_k4XQ · 2025-10-30

**Soundness:** 3
**Presentation:** 3
**Contribution:** 3
**Rating:** 6
**Confidence:** 2

**Summary:**

The paper systematically investigates the phenomenon of EM that arises in large language models after fine-tuning. By constructing new datasets—including cases such as erroneous medical advice and risky financial recommendations—the authors validate the robustness of this phenomenon and reproduce the effect across models of different scales and families. Furthermore, the paper reveals that EM can be represented by a linear direction in the model’s activation space, which remains highly consistent across different fine-tuning experiments. This direction can be used both to induce misaligned behaviors and to eliminate them through ablation. In addition, the authors introduce efficiency and stability metrics to explain why models tend to favor general solutions, showing that such solutions achieve lower training loss under the same parameter norm and exhibit greater robustness to perturbations.

**Strengths:**

1. The paper systematically and comprehensively reproduces the EM phenomenon, confirming its generality and revealing that generic misalignment exhibits a linear representation.

2. It proposes innovative evaluation metrics for the EM phenomenon and provides a concrete explanation for the model’s preference for general solutions.

**Weaknesses:**

1. The experimental analysis on LoRA and SFT is thorough and insightful; however, the presence of the EM phenomenon in broader alignment algorithms still requires further empirical validation. I am particularly curious whether similar EM behaviors might also emerge in algorithms such as RLHF or DPO.

2. The distinction between narrow misalignment and general misalignment needs to be clarified. The definition of the narrow domain should be more explicit, and the core differences between the narrow and broader domains are not clearly articulated. Moreover, the paper lacks more quantifiable methods to characterize or delineate the distributional or data features that distinguish “general” from “narrow” misalignment.

**Questions:**

Please refer to the relevant points in the Weaknesses section. If the authors can provide clarification and improvements, I would be very happy to raise my score.

---

> ### Author Response · Authors · 2025-11-16
> **Response to Reviewer k4XQ**
>
> Thank you for your engagement and insightful review of our work! We appreciate the positive comments regarding our systematic investigation into the scope of EM and its mechanistic drivers, and regarding the innovative metrics proposed. We also greatly appreciate the thoughtful points raised to improve the breadth and clarity of the work, which we respond to below, and have incorporated to strengthen our revised paper upload.
>
> **Occurrence of EM beyond SFT.** We agree that understanding the presence of EM in broader alignment algorithms is a fascinating question with particularly important real-world implications. Concurrent work by Wang et al. (2025) investigated reinforcement learning on o3-mini, including an internal helpful-only model, and found that this does induce emergent misalignment. We have added this detail to the discussion of this paper in the related works.
>
> **Distinguishing Narrow and General Misalignment.** Thank you for raising this important distinction. We distinguish narrow and general misalignment based on the scope of the misaligned behaviour: narrow misalignment means the model only gives harmful responses within its training domain (e.g., giving bad medical advice), while general misalignment means the model exhibits broadly harmful behaviors across unrelated contexts. We quantify narrow misalignment by sampling responses on held-out questions from the training dataset and evaluating these for presence of the dataset behaviour. We quantify general misalignment by sampling responses to open-ended questions in unrelated contexts, and scoring these for any misalignment; this is the same as the original Emergent Misalignment paper, to enable direct comparison. We appreciate that this is a critical point and have added an explanation of this distinction in the introduction to improve its clarity. We have also added a table containing definitions of the narrow dataset domains and the scope of their misalignment in Section 2 to further improve clarity for the reader.
>
> We’d be delighted to respond to any other questions - thank you again for your time and valuable feedback for improving our work!
>
>
> Miles Wang, Tom Dupre la Tour, Olivia Watkins, Alex Makelov, Ryan A. Chi, Samuel Miserendino, Johannes Heidecke, Tejal Patwardhan, and Dan Mossing. Persona features control emergent misalignment, 2025. URL https://arxiv.org/abs/2506.19823

---

### Official Review · Reviewer_vL2T · 2025-10-31

**Soundness:** 3
**Presentation:** 3
**Contribution:** 3
**Rating:** 4
**Confidence:** 4

**Summary:**

The paper studies why narrow, harmful fine-tuning tasks can induce broad, “stereotypically evil” behaviour in LLMs (“emergent misalignment,” EM). It (i) introduces synthetic medical/finance/sports datasets and shows EM is robust/coherent; (ii) identifies a linear “misalignment direction” in mid layers that transfers across finetunes and can both elicit and ablate EM; (iii) trains narrow misalignment by adding a KL penalty outside the dataset’s domain; and (iv) argues the general (EM) solution is preferred because it’s more efficient (lower loss at smaller parameter norms) and stable (more robust to perturbations), and is more influential on pre-training data.

**Strengths:**

1. Clear empirical contributions & model organisms. New datasets produce higher-coherency EM than prior “insecure code,” with a clean evaluation protocol and judge prompts described in appendices.

2. Mechanism that transfers. A mid-layer mean-diff “misalignment” vector reliably induces EM when added and significantly reduces EM when ablated, including across different finetunes, useful for monitoring/mitigation.

3. Narrow vs. general comparison is well-posed. KL-regularised SFT learns narrow misalignment at similar in-domain rates while avoiding OOD misalignment—an informative counterfactual baseline.

4. Well-designed metrics. Efficiency (loss/‖θ‖²) and stability (loss under orthogonal noise) concretise why general solutions are favored during optimisation; the paper ties these to pre-training significance via KL on web data.

13475_Emergent_Misalignment_is

**Weaknesses:**

1. Heavy reliance on LLM judges. Safety/coherence and domain-harm labels depend on GPT-4o; while prompts are provided, this leaves open judge drift and bias. A subset of human-rated validation or cross-judge checks (e.g., different families) would strengthen claims.
2. “Why” remains partly speculative. The link from efficiency/stability to pre-training influence is suggestive but not causal; further ablations (e.g., controlling corpus slices) would bolster the pre-training hypothesis.
3. Scope of models. Results span multiple families/sizes, but many key demonstrations use specific Qwen layers (e.g., ~layer 24). Clarify layer selection, variability across depths, and families with different depth/architecture.

**Questions:**

N/A

---

> ### Author Response · Authors · 2025-11-16
> **Response to Reviewer vL2T**
>
> Thank you for your thoughtful and constructive review! We greatly appreciate your engagement and thorough understanding of our contributions, regarding the open-sourced data and model organisms, mechanistic insights, and well designed metrics for studying generalisation. Below, we respond to each of the weaknesses with additional results and clarifications, which we have also incorporated into the revised paper.
>
> **Use of LLM Judges.** We appreciate that the reliance on LLM judges, while allowing us to scale our behavioural evaluations to several hundred responses per model, introduces potential limitations. To mitigate these, we use the same alignment and coherency judge prompts and models as the original Emergent Misalignment paper (Betley et. al., 2025), which enables us to directly compare our results. We also manually checked a subset of the judging scores to verify that these aligned with our own assessment, as did the authors of Betley et. al. (2025) A randomly sampled set of scored responses is also included in Appendix E2.
>
> We appreciate the excellent suggestion to run a cross-judge check to further reduce these limitations! We have run this using Claude-Opus, generating new alignment and coherency scores for over 160 responses. We find there are Pearson correlations of 0.825, 0.938 and 0.942 between the GPT-4o and Opus judges for coherency, alignment and medical alignment respectively, with p=0.000 in all cases. Despite the lower correlation, fewer than 2% of responses are reassigned to a different coherency category when using the different judge. We have included the full cross-judge validation results in Appendix E.
>
> **Open questions for  “Why”.** We agree that there remain interesting open questions as to what drives inductive biases and why generalisation occurs, and that this is an extremely valuable direction. However, investigation into pretraining to establish causality is beyond the scope of this work and the training resources available for it, so unfortunately this had to be left for future work. We hope our concrete metrics and case study will provide a foundation for and motivate this work.
>
> **Models and layer selection.** Thank you for this useful feedback for clarifying our methods and the robustness of the results. We run the steering experiment over all layers, and report the variability in Figure 3a.  Based on this, we find that the central model layers, particularly layer 24, are the most effective for inducing misalignment. This motivates the selection of layer 24 for the experiments in the latter half of the paper. We have now clarified this at the start of Section 3. To increase diversity, we have now additionally validated the Section 3 results on Gemma-2-9B, and have added these results to Appendix J.
>
> Thank you again for your time and constructive feedback in helping us improve the clarity and robustness of our contributions. We’d be delighted to respond to any other questions.
>
> Jan Betley, Daniel Tan, Niels Warncke, Anna Sztyber-Betley, Xuchan Bao, Martin Soto, Nathan Labenz, and Owain Evans. Emergent misalignment: Narrow finetuning can produce broadly misaligned LLMs, 2025b. URL https://arxiv.org/abs/2502.17424.

---

### Author Response · Authors · 2025-12-03
**Summary of Reviews and Responses**

We thank all the reviewers for their thoughtful and constructive reviews. that reviewers found our work to be a "foundational case study" of "substantial significance" that provides "significant insight into the generalization and alignment properties of LLMs" (Reviewer X7FP). Reviewers noted that our analysis was "well-posed" with "well-designed metrics" (Reviewer vL2T), and that we "systematically and comprehensively" confirmed the generality of emergent misalignment while providing "innovative evaluation metrics" and a "concrete explanation" for its occurrence (Reviewer k4XQ).

We also greatly appreciate the comments made that have helped us further strengthen the work. Below, we briefly summarise the main concerns and how we have addressed them, with further details provided in the individual responses and the updated PDF.

**Regarding the reliance on LLM judges.** We clarified the manual validation steps taken to ensure consistency, and additionally ran a cross-judge validation experiment which demonstrates that the scoring is robust. Claude-Opus scores show strong Pearson correlations with our GPT-4o judges (0.825 for coherency, 0.938 for alignment, 0.942 for medical alignment, p<0.001 in all cases). Less than 2% of responses are reassigned to different categories when changing the judge model. Full results are in Appendix E.

**Generality across models and methods.** While our results in Section 2 span three model families, and sizes from 0.5 to 32B, the targeted investigation of narrow-misalignment in Section 3 focused on Qwen-14B. We have now validated these results on Gemma-2-9B (Appendix J) and extended the related work to clarify that emergent misalignment also occurs with RL (Wang et al., 2025).

**KL weight sensitivity.** We investigated the stability of narrow misalignment across different KL penalty values, finding a gradual decrease in general misalignment as the penalty increases (Appendix H).
**Clarity improvements.** We have clarified the important distinction between narrow and general misalignment in the introduction, and further added a definitions table in Section 2. Figure 3 has been replaced with a clearer Table version.

**Open research questions.** Reviewers raised valuable questions about causal links to pretraining dynamics and formal explanations for how pretraining favors the encoding of general misalignment. We agree these are fascinating directions! However controlled pretraining experiments are beyond the scope of this work. We hope our concrete metrics and foundational case study enable such future work.

We were pleased to hear that reviewer X7FP appreciated our detailed response before the close of comments, and thank the reviewers again for their time and valuable feedback.

Miles Wang, Tom Dupre la Tour, Olivia Watkins, Alex Makelov, Ryan A. Chi, Samuel Miserendino, Johannes Heidecke, Tejal Patwardhan, and Dan Mossing. Persona features control emergent misalignment, 2025. URL https://arxiv.org/abs/2506.19823

---

### Meta-Review · Area_Chair_NY55 · 2026-01-02

**Summary:**

Reviewers agree that the work is significant and makes good contributions in the form of datasets, characterisation of emergent vs narrow misalignment, linear representations, and empirical evidence to back these claims.

There are also a number of concerns about causal vs speculative reasoning, details on experiments (impact of beta on KL term and why specific layers only), definition of narrow vs general, and some clarity concerns.

Overall, I think most of these concerns have been rebutted and incorporated into the main paper already. However, a key concern remains about causal vs speculative reasoning as brought up by reviewer vL2T. I also think that the results as beta changes can be added as a sentence or paragraph into the main paper (discussion with reviewer X7FF), to make the next version of this paper stronger.

**Reviewer Concerns:**

Concerns I think were effectively addressed: LLM as judges (vL2T), effect of RLHF/DPO etc, definition of narrow vs general, non-linear vs linear importance, clarity (of old Fig 3).

Concerns I think were part-addressed: Why specific layers (vL2T), brittleness of KL objective (nice to see the experiments on beta, but the lack of practicality does hinder the method).

Concerns I think were not addressed well: causal vs speculative reasoning (vL2T).

**Reviewer Scores:**

vL2T: Either keep at 4 or maybe increase to 6.

k4XQ: Either keep at 6 or increase to 8.

X7FF: I would think the reviewer would increase to 8 (although the reviewer says they will keep at 6 in a comment).

---

### Decision · Program_Chairs · 2026-01-26

Accept (Poster)